# More Timber in Construction: Unanswered Questions and Future Challenges

**Jim Hart and Francesco Pomponi *** 

REBEL—Resource Efficient Built Environment Lab, School of Engineering and the Built Environment, Edinburgh Napier University, 10 Colinton Road, Edinburgh EH10 5DT, UK; j.hart@napier.ac.uk
* Correspondence: f.pomponi@napier.ac.uk

**Abstract:** The built environment is one of the greatest contributors to carbon emissions, climate change, and to the unsustainable pressure on the natural environment and its ecosystems. The use of more timber in construction is one possible response, and an authoritative contribution to this growing movement comes from the UK's Committee on Climate Change, which identifies a "substantial increase in the use of wood in the construction of buildings" as a top priority. However, a global encouragement of such a strategy raises some difficult questions. Given the urgency of effective solutions for low-carbon built environments, and the likely continued growth in demand for timber in construction, this article reviews its sustainability and identifies future challenges and unanswered questions. Existing evidence points indeed towards timber as the lower carbon option when modelled through life cycle assessment without having to draw on arguments around carbon storage. Issues however remain on the timing of carbon emissions, land allocation, and the environmental loads and benefits associated with the end-of-life options: analysis of environmental product declarations for engineered timber suggests that landfill might either be the best or the worst option from a climate change perspective, depending on assumptions.

**Keywords:** structural and engineered timber; life cycle assessment; harvested wood products; carbon storage; built environment; climate change mitigation

## 1. Introduction and Background

As the energy efficiency of new buildings improves, driven in part by tightening regulations [1], the carbon emissions arising from their operation should be lower than their forerunners. As a result, it is now widely recognised that the environmental costs—particularly the embodied carbon—of materials incorporated in a building represent an increasingly significant proportion of the building's life cycle environmental burden. There is now more interest in this issue from professional organisations [2], policy-makers [3], and from people and organisations wanting to signal their attention to climate change.

Mineral-based construction materials, such as metals, cement, and glass, are often characterised as coming from finite resources and requiring significant inputs from fossil fuels to process them into the sophisticated products required by the industry [4]. Timber, by contrast, is presented as a 'renewable' material (and therefore, by subtle implication, not 'finite'—albeit over an infinite timespan) that comes with an integrated fuel supply. Additionally, timber construction products physically embody carbon that, prior to the trees' intervention, existed in the form of atmospheric carbon dioxide ($CO_2$). Thus, it can be argued that the combined forestry, wood products, and construction systems can play a role in carbon sequestration strategy to mitigate climate change [5,6]. The carbon physically stored in the timber in this way is different from the 'embodied carbon' widely discussed in the literature and in the rest of this article. Embodied carbon (EC), here, refers to the greenhouse gas (GHG) emissions associated with the production of a material or product.

The Committee on Climate Change (CCC) in the UK [7] makes the case for an increase in land area under sustainably managed forestry, which can lead to an increase in the carbon stored in the forests and in durable harvested wood products (HWP). It argues that biomass should only be harvested at a sustainable rate and that such material should be put to the most climate-beneficial use: a "substantial increase in the use of wood in construction" is identified as a priority [6]. It notes the importance of construction timber in GHG emission abatement, by storing carbon and by displacing materials with higher EC. Construction timber in the UK results in over 1 Mt (1 megatonne) of additional $CO_2$ being stored per year in new homes, with potential to increase to 3 $MtCO_2$ per year by 2050, with similar progress in the commercial and industrial sectors through uptake of new types of engineered timber systems [7].

The CCC analysis assumes a continuing shift away from masonry towards timber-frame houses, and the analysis includes biogenic carbon storage. This is justifiable in the context of the report—which is about identifying the best role for biomass in a low carbon economy—but the question of whether it is justifiable to produce more biomass to meet such needs opens up questions discussed here. Furthermore, increased use of engineered timber such as cross-laminated timber (CLT) will result in demand for more construction timber per unit of floor area in comparison to more conventional timber-frame construction. It is important to explore the environmental costs and benefits of such a shift.

It is generally understood that trees—and therefore afforestation and reforestation—make a huge range of contributions to the viability of all forms of life on the planet. The process of sequestering and storing carbon from the atmosphere is just one facet of this, and the great global potential for afforestation/reforestation to mitigate climate change has been noted [8]. On the other hand, it has also been argued that terrestrial carbon dioxide removal through photosynthesis cannot prevent large temperature rises without eliminating virtually all natural ecosystems, and that dramatic emissions reductions are still the priority. There is much intellectual ground to explore between these two points of view [9].

## 1.1. Life Cycle Assessment (LCA)

LCA can be used for the scientific investigation and reporting of the environmental impacts of buildings, products, and materials. A range of impacts can be modelled, but this article focuses on the climate impact indicator GWP100 (Global Warming Potential, 100-year horizon) [10]. LCA carried out according to EN 15978 [11] compiles an inventory of resource flows required for the creation of a unit of material or functional unit of a product (e.g., 1 $m^3$ oven-dry timber, or 1 $m^2$ wall designed to a particular specification), and then assesses the impacts. The life cycle stages are shown in Table 1.

Suppliers to the construction industry provide LCA information in the form of Environmental Product Declarations (EPD) based on Product Category Rules (PCR) in accordance with the International Standard ISO 14025 (BRE, 2014). EPD, in turn, provide background data for LCA studies of buildings. The LCA underpinning an EPD will usually be 'cradle-to-gate, with options,' i.e., including modules A1–A3 as a minimum. A cradle-to-grave assessment gives a more complete picture, but at the cost of greater uncertainty in relation to assumptions about the future. Even a cradle-to-grave assessment misses key differences between renewable and mineral products, as these are taken to be outside of the product system and are covered by module D, or outside of the anthropogenic system altogether.

LCAs follow methodological conventions or face methodological choices that can have an impact on the relative rating of different materials. For instance, whilst the scope of module A1 includes the forestry management inputs during the many years from site preparation through to harvest, it does not include assessment of carbon stocks and flows in the soil, nor the flux of methane and nitrous oxide. Nor does the assessment consider the consequential carbon cost of harvesting as opposed to—for instance—allowing the trees to stand for a further period before harvesting. Although the temporary carbon storage property of timber is central to the EPD of timber products, it is not generally included in LCAs of buildings except as supplementary information.

**Table 1.** Life cycle stages categorised according to BS EN 15978. A full account of embodied carbon—cradle to grave—should include all carbon emissions attributed to A1–C4, but excluding B6–7). In practice, many studies are more limited.

| A1–A3 | | | A4–A5 | | Building Life Cycle. B1–B7 | | | | | C1–C4 | | | | D |
|---|---|---|---|---|---|---|---|---|---|---|---|---|---|---|
| Product Stage | | | Construction Process Stage | | Use Stage | | | | | End-of-Life Stage | | | | Supplementary Info—Beyond the Building Life Cycle |
| A1 | A2 | A3 | A4 | A5 | B1 | B2 | B3 | B4 | B5 | C1 | C2 | C3 | C4 | |
| Raw material supply | Transport | Manufacturing | Transport | Construction | Use | Maintenance | Repair | Refurbishment | Replacement | Deconstruction demolition | Transport | Waste processing | Disposal | Benefits and loads beyond the system boundary: Reuse, recovery, recycling |
| | | | | | | | B6 Operational energy use B7 Operational water use | | | | | | | |

The use phase—stage B—is usually pared down or neglected altogether in building LCAs [12]. This is because design for identical performance is tacitly assumed (although this will be unrealistic in many cases) and also because studies explicitly focusing on EC will exclude operational carbon emissions by definition. Poor data availability on maintenance and repair regimes and on the probability of major refurbishment is also a factor limiting the scope of many studies.

A limitation of LCA of buildings and construction products is our inability to predict the fate of buildings and components under future ownership, and at end of life (EoL), which might be decades or centuries away whatever the original design intent. A further problem is that impacts far into the future are modelled with reference to today's technologies and practices, resulting in systematic bias.

Excepting some disposal routes for timber, the stage C impacts are likely to be reported as having relatively low values compared to the cradle-to-gate impacts of modules A1–3. Pomponi and Moncaster [13] found examples in the academic literature whereby the stage C impact is approximated with a small fixed percentage of the EC of the product stage. Any such assumption concerning engineered timber structures should be avoided, as the ecoinvent database indicates C4 EC coefficients relating to landfill that are very close in value to those for the product stage (e.g., for CLT A1–A3: 0.55 kgCO$_2$e/kg; C4: 0.54 kgCO$_2$e/kg—GHG emissions in terms of kg of CO$_2$ equivalent per kg of CLT).

*1.2. Circular Economy (CE)*

The concept of a Circular Economy (CE) has also gained traction in recent years, and much of the discussion around the sustainability of material resources is now framed in these terms. Advocates in the UK include Zero Waste Scotland and the Ellen MacArthur Foundation [14,15]. There is no single definition, but the essential point is that economic growth should be decoupled from resource extraction. This can be advanced by keeping goods and materials in circulation for longer periods and using them more intensively and productively. Recycling is seen as something of a failure, as it involves a loss in value from manufactured goods. Instead, the emphasis should be on durability, maintenance, reuse, remanufacture, and the business models needed to facilitate these strategies.

CE presents challenges for all types of structural products, and wood is no exception. The reuse of structural steel is possible, but is hindered by current regulations and a poor financial case [16,17]. Reuse of timber structural elements presents similar challenges, plus the additional conundrum that engineered structural timber is typically a composite of organic and mineral substances that may be impossible to separate in a controlled way. This raises the question of whether to store the materials ad infinitum (e.g., in landfill, which is anathema to CE-based policy) or to return the organic component to the biosphere (through combustion for instance), with the associated release of pollutants or costs of additional pollution control measures.

## 2. LCAs of Buildings Using Timber Structures

Different types of construction systems have been much compared through LCA, with a focus primarily on steel and masonry systems, but with increasing interest in timber.

Different approaches have been adopted for investigating the EC of buildings with different characteristics. These include like-for-like comparisons of pairs of buildings that differ only in the aspect of interest (e.g., the structural material), to sweeping searches for benchmarks and trends from large samples of buildings differing in function, location, scale, and very often in the methodologies and scopes of the studies.

Several studies have found significantly lower values for the EC in timber structural frames than concrete or steel counterparts, without including biogenic carbon content in the account [18–20]. A curiosity associated with one of these [20] is that when extending the assessment to include biogenic carbon according to the Cherubini method [21], the carbon cost associated with the use of biomass energy outweighed the carbon benefit associated with storage, and the relative advantage of timber was actually reduced. Although concrete has lower EC than timber per unit of mass [13], there is much evidence to suggest that the use of timber results in buildings with lower EC [22].

In an assessment of functionally equivalent concrete and timber designs for a small road bridge in Sweden, the timber bridge had a 22% lower EC [23]. Taking a similar approach to residential buildings, a cradle-to-grave LCA result for a CLT-based home had 42% lower EC than its concrete counterpart [24]. In both studies, the superiority of the timber variant was confirmed within parallel assessments using dynamic LCA.

Some researchers have attempted to generalize the emissions benefits from substituting wood in for conventional construction materials. In their meta-analysis, Sathre and O'Connor [25] found that, on average, for every tonne of wood used in construction, 3.9 tonnes of $CO_2$e emissions are avoided, providing a rationale for substituting wood for other products. After converting to the equivalent units, this is towards the lower end of the range of displacement factors identified by Geng et al. [5] for construction timber, which is 0.25–5.6 kgC substituted per kgC in the timber. The wide range is a reflection of the different contexts in which timber is compared to other construction materials, and also to differences in the scope of the assessments.

The balance of the evidence discussed above points towards timber comfortably being the lower carbon option when modelled through LCA, certainly in cradle-to-gate analysis, and probably cradle-to-grave too, without relying on arguments around carbon storage.

## 3. LCA limitations, Omissions, and Variations

Many aspects of the life cycle of construction timber are either poorly characterised or neglected altogether in typical LCA practice. This stretches from carbon fluxes and stocks in forest soils to the eventual decay or destruction of the HWP. Key areas of neglect or uncertainty include the assessment of carbon stocks in the forest and HWP, nonanthropogenic GHG fluxes in the forest, competition for land, and the choice and assessment of EoL scenarios.

### 3.1. End of Life

A detailed understanding of life cycle carbon emissions requires an understanding of the eventual fate of the materials involved.

The main options for EoL structural timber are illustrated in Figure 1.

Listed in 'waste hierarchy' order and with reference to the labels in Figure 1, the EoL options—at least in theory—are:

1.  Reprocess/reuse. Complete sections of timber might be salvaged for reuse in a different building. Stored carbon continues to be stored, and the LCA costs and benefits of the counterfactual virgin timber products in that next building are substituted.
2.  Recycle. Timber is chipped or shredded and turned into boards or animal bedding. Again, stored carbon continues to be stored, but for a shorter expected period than for reuse as structural timber. In practice, this is not a realistic option for post-consumer timber arising from demolition.
3.  Energy Recovery. Timber is burned in an energy recovery facility. In this case, the stored carbon is returned to the atmosphere as $CO_2$ along with much smaller quantities of GHGs methane and nitrous oxide. On the other side of the equation, the heat and/or power harnessed for other uses offsets GHG emissions from other systems.
4.  Managed landfill. In this case, a high proportion of the carbon is stored for the long term, but the decomposition of the cellulose and hemicellulose that does occur produces landfill gas (LFG), which is typically around 50% methane, a powerful GHG. The methane will—in some proportion—either be captured and burned with energy recovery (producing an offset), captured and flared (no offset), or leak into the atmosphere. The impact of landfilling timber is also dictated by the rate and extent of its decay, parameters that are difficult to calculate given the variation of contexts in which landfills exist (e.g., climates and management practices), and the dearth of studies of degradation, in situ, over periods of decades [26]: analysis of wood samples excavated from Australian landfills being a notable exception [27].

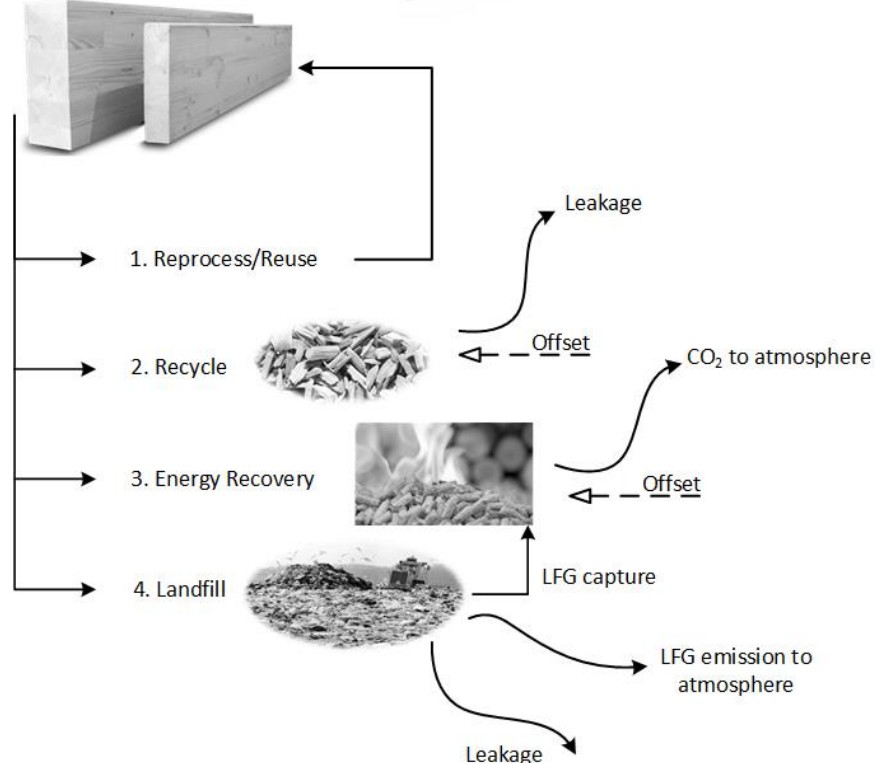

**Figure 1.** Simplified diagram of end of life options for a timber construction element (illustrated by glulam beams). Solid arrows represent material and carbon flows. Dotted arrows represent carbon accounting offsets relating to the counterfactual (what would have happened in a related system if the timber had not, for instance, been recycled or consumed in an energy recovery facility). 'Leakage' occurs in almost any process, involving some combination of material losses and carbon cost arising from resource inputs. LFG: landfill gas.

It is rarely in the gift of building design teams to specify how and when buildings will end their lives, and this uncertainty is compounded by the complexities associated with the calculation, reporting, and interpretation of carbon emissions over a long timeframe.

Beyond the System—Including Stage D in the Analysis

Module D can be used to report results 'beyond the life cycle' as a supplement to the LCA itself. Adding module D figures to those of the product LCA is not generally advised, but given that stage C and stage D both need to be assessed in order to reach an informed view about EoL pathways, approached with suitable caution, combining them into a single figure can be useful in helping us to navigate EoL towards the lowest impact path. To this end, we have studied a selection of publicly available LCAs [28–33]—mostly EPD—for glulam and CLT. The ranges of values for C + D for each pathway are indicated in Figure 2. The wide ranges and the extent of overlap between the different EoL pathways is at odds with the straightforward assumptions about EoL implicit in CE-thinking and demands explanation: is this 'real' or is it a result of the conventions underpinning the EPD?

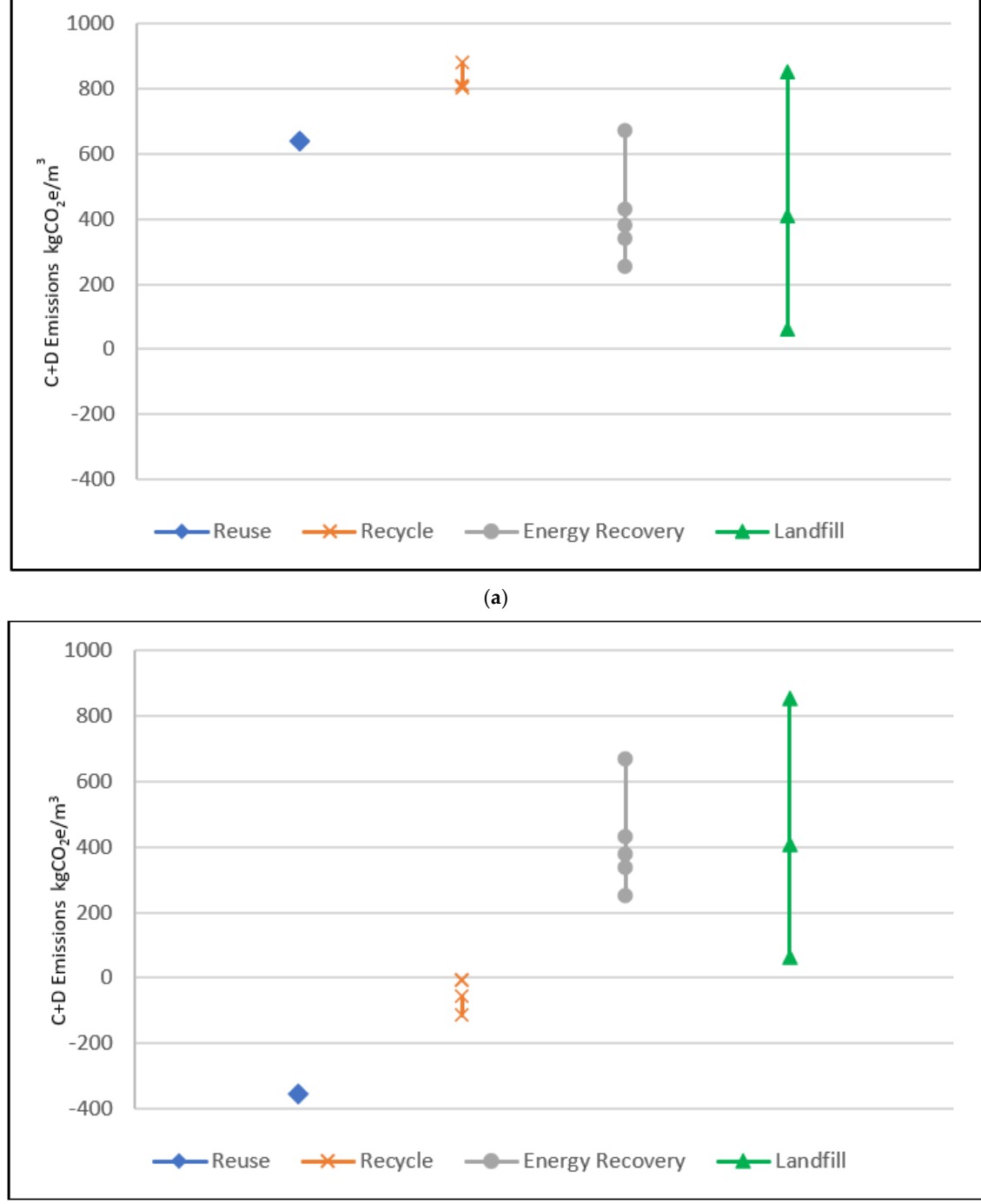

**Figure 2.** EoL GWP figures from a selection of LCAs. The lines indicate the range between maximum and minimum where there is more than one datapoint in the set. (**a**) C + D stages, as reported in the LCAs, apart from the reuse figure which has been adjusted downwards as the biogenic carbon was double-counted. (**b**) The same data reinterpreted from a waste management perspective. EoL: end of life; GWP: Global Warming Potential; LCA: Life Cycle Assessment.

The reuse and recycling figures are derived from two main elements. The first is the biogenic carbon (reported as a credit in these documents at stage A) being transferred to the next product system when material reaches an 'end-of-waste' state at C3, which equates to a debit of approximately 800 to 1000 $kgCO_2e/m^3$ (depending on the properties of the particular product). The next product system

continues to store the carbon, but it is no more than would be stored in the counterfactual case reliant on virgin timber. The second is a smaller credit covering the fossil fuel inputs that would otherwise have been required in the secondary product system to produce a functionally equivalent product from primary materials: 380 $kgCO_2e/m^3$ in the case of reuse, but much smaller values (8 to 142 $kgCO_2e/m^3$) for recycling, where the displaced material is chipped or shredded virgin timber as opposed to a fully formed construction product. As a result, the data suggests—as would be expected—that reuse should be preferred to recycling. The missing item from this analysis, however, is that from an EoL waste management perspective, apart from any material that is lost, the biogenic carbon stored in the HWP is not emitted back into the atmosphere: the transfer to the next product system is carbon neutral (disregarding any material losses in the reuse process), and the displacement of a new product from virgin timber is also neutral (over time) with respect to biogenic carbon. From this perspective, the figures for reuse and recycling should be below the axis (Figure 2b), which is an important finding, as taking the EPD at face value could serve to undermine CE by discouraging design for reuse.

The key to the wide range of figures reported for the energy recovery scenarios is the counterfactual energy system (which is not always detailed). In all cases, the biogenic carbon leaves the product system (being emitted directly to the atmosphere) and is debited accordingly, and a credit is applied reflecting the substitution benefit of the energy provided. The LCA reporting the highest such offset states that the counterfactual would be thermal energy from natural gas. At the end of the product life—perhaps fifty years from now—one might assume that the combustion of natural gas without carbon capture and storage is no longer an option and is therefore an unrealistic assumption. However, the approach complies with EN 15804:2012 6.4.3.3 [34], which requires assessment based on current average technology or practice and also has the advantageous side effect of helping us to evaluate options for materials reaching EoL now. The lowest value reported for energy recovery in module D is likely to arise from the large contribution of hydropower to meeting the existing energy needs in the local context.

In the case of landfill, only a fraction of the biogenic carbon leaves the product system: the remainder, which is in timber that does not decay within the time horizon of the LCA, continues to be stored in the system: in the forthcoming version of the standard (EN 15804:2012 + A1:2013/FprA2), a 100-year cutoff is expected to be introduced, after which point any carbon remaining in landfill is deemed to have left the system. The numbers on landfill reported in Figure 2 therefore primarily relate to LFG production, with a debit for its emission, and a smaller credit for whatever proportion is utilised. Widely varying estimates of the extent of timber degradation and LFG production and utilisation are used in the LCAs. In the Australian example [33], two landfill decomposition scenarios were considered, with degradable organic carbon fractions (DOCf) of 0.1% and 10% assumed, i.e., two orders of magnitude apart, based on different studies. The Wood for Good LCA, by contrast, assumes organic carbon conversion at a rate higher than either figure, in line with standard modelling used in the UK [35]. Negligible carbon losses have been recorded from samples recovered from one Australian landfill after several decades, but losses (DOCf) were around 18% in another [27].

It is clear that if a slow rate of decomposition of timber in landfill is justified, for engineered timber that is currently reaching EoL, landfill might be a better option than energy recovery, despite being banned or constrained in some legislations. More evidence is still required to demonstrate the differential rates of decay of different types of organic material in landfill and identify the conditions that allow for stable storage of timber over the long term, however.

For timber that is not destined to reach EoL for several decades, it would be useful for LCAs to present alternative scenarios that consider future technologies (and future penetration of existing technologies). In such cases, the C + D figures for energy recovery might look less favourable. In one set of scenarios investigated, recycling of end-of-life wood into new products has been found always to be better than burning or burying, with burying better (from a climate perspective) than burning to displace gas boiler fuel [36].

Modules C and D should be seen as tools to facilitate a transparent investigation of possible alternative EoL scenarios, but typically they are not used in this way, as little attempt is made to justify assumptions and analyse sensitivity to them. Furthermore, it can be difficult to interpret the results for recycling and reuse scenarios. Ultimately, beyond its 'first' life in a building, timber disposal and use results in a range of positive and negative effects, including fossil fuel displacement, carbon storage, decay, and methane leakage. Wider use of timber will inevitably increase the relevance of all of these beyond-EoL scenarios, and, as such, they deserve to be better understood.

### 3.2. Evaluation of Temporary Carbon Storage

For short-lived mineral or synthetic products, most carbon emissions occur within a short space of time during the months leading up to the product. For long-lived products such as buildings, with a biogenic element, time is a more important variable, as relevant carbon fluxes occur in the forest decades before and after construction, and from the eventual destruction or decay of the HWP. As such, these issues merit deeper consideration.

### 3.2.1. LCA and GWP Time Horizons

By convention, the climate change impact of a single-pulse emission of GHGs is distilled down to a single figure, usually GWP100 for a building LCA. GWP100 is not a direct measure of climate impact, as it represents the cumulative radiative forcing over the subsequent 100 years, relative to the radiative forcing caused by a pulse emission of 1 kg of $CO_2$ [37]. There is not a linear relationship between this and the physical endpoints in the system such as temperature change or sea-level rise. GWP100 allows emissions consisting of different gases with different interactions with infrared radiation and with different decay rates in the atmosphere to be described in terms of a single figure of kilograms of carbon dioxide equivalents (kg$CO_2$e). However, the choice of time horizon is, in a sense, arbitrary, and it affects the result. For instance, choosing a shorter time horizon would increase the reported relative significance of any methane emitted, with respect to $CO_2$. Balcombe et al. [37] reviewed other climate metrics that are available and made recommendations on their application, which include transparency, reporting the impacts associated with different gases separately in some circumstances, and using dynamic approaches and end-point metrics for assessment of long-term decarbonisation pathways.

Similarly, the time horizon chosen for the assessment itself is a loaded choice and effects the results, as emissions occurring after the time horizon are excluded, potentially an important consideration in the context of long-lived buildings and landfill sites.

### 3.2.2. Accounting for Biogenic Carbon Storage

As mentioned above, cradle-to-gate EPD for timber products report negative GHG emissions on account of the biogenic carbon stored in the product. Businesses have been known to take this as a license to claim that their timber-heavy projects are also carbon negative, implying that their time horizon is less than the projected lifetime of the building. The conservative approach to assessing biogenic carbon in building LCAs is to assume that the biogenic carbon is stored only temporarily, reverting to the atmosphere within the timeframe of the assessment, and therefore has no overall impact. So, no credit is given for the temporary carbon storage on the timescale of a typical building lifespan, nor for long-term storage (hundreds or thousands of years) in cathedral-like buildings, or in very dry or permanently waterlogged and anaerobic conditions below ground where carbon can be stored for millennia: Blanchette [38] provides an interesting review from an archaeological perspective.

One method for accounting for awarding credit for temporary carbon storage is offered by PAS2050 [39], which effectively postpones part of the impact of delayed emissions to a period beyond the assessment period, and therefore beyond the system. It has been observed that such methodologies tend to overstate the benefit of temporary carbon storage and conflict with the precautionary approach that underpins LCA [40].

In any parts of the world that include increasing stocks of growing boreal or temperate forest, such as Europe at least for the time being [41] and increasing stocks of HWP, an associated climate benefit surely exists. Arguably, the global stock of stored carbon can continue to grow for the foreseeable future, through a process of continuing growth in global afforestation, material substitution by biogenic materials, and long-term carbon storage in products and landfills, although there is an argument that the best place to store biogenic carbon is in a living tree [42]. However, anything that adds to the total stock of stored carbon is providing an environmental service that, ideally, would be captured by LCA.

A system of 'Dynamic LCA' has been developed to bring a more scientific approach to bringing the timing of emissions into scope: the timing of all biogenic carbon fluxes can be included in the life cycle inventory, allowing carbon storage in trees and soils, for instance, to be factored into the assessment [43]. The quality of data available is an obvious constraint to this, and results are impacted by the choice of whether the tree attributed to a product is the tree harvested or the tree planted to replace the one that is harvested. Or, put another way, does the sequestration precede the emissions associated with production (good) or follow on from the production (less impressive)? We would suggest that the most useful approach would be to assume that the various agents responsible for the production of—for instance—a timber house can most strongly influence the processes that happen at around the time of production, and should not take credit for events (planting of trees) that occurred decades earlier. Fouquet et al. [44] make the same choice, for related reasons, in their assessment of a house design in alternative timber and concrete configurations using dynamic as well as conventional LCA. The conventional approach—leaving out biogenic carbon and module D—shows the timber option to have at least 10% lower EC than the concrete options. Using the dynamic approach, the advantage for timber becomes considerably more marked ten years or so into the life of the building (when carbon sequestration by the new trees is starting to make an impression) and projected centuries into the future.

Whilst there is some consensus that carbon stored in HWP such as structural timber has a role to play in mitigating climate change, there is less consensus on how this role should be evaluated, and so current thinking tends towards leaving this out of LCA. The assessment of biogenic carbon remains a field where consensus is yet to be reached, since different methodological choices and assumptions lead to opposite conclusions and incorrect or inaccurate assessments of biogenic carbon can be the cause for missed opportunities as well as inefficient or counterproductive strategies [45].

### 3.2.3. Carbon Pools in Forests, HWP, and Landfills

Many authors have modelled the combined forestry-HWP system to gain an improved understanding of how carbon is removed from the atmosphere, moves between pools (trees, soils, buildings, and landfill) while it is being stored within the system, before it is ultimately returned to the atmosphere. Such models should take account of the fact that not all GHG fluxes are $CO_2$; many also include 'virtual' carbon pools, which represent the emissions avoided as a result of using HWP instead of other products. Such virtual pools frequently account for most of the carbon in the system. For instance, Knauf [46] found that the climate change mitigation contribution of Germany's forest sector amounts to as much as 15% of Germany's total GHG emissions: more than half of this is related to material and fuel substitution, with the HWP stock accounting for only 6.5% of the total. However, substitution has been labelled merely a theoretical carbon pool that is overestimated by some authors by an order of magnitude, and with a built-in double-counting mechanism that is initiated when the HWP is itself replaced [47]. Some models accumulate the benefits of substitution almost ad infinitum, whilst others define a more limited period of interest, such as 70 years [48].

Including substitution benefits in the model can favour more intensive forestry management practices, greater inputs and higher rotation rates, and all the environmental impacts that go with these. In their regional model, Gustavsson et al. [49] found that their most intensive scenario was best from a climate perspective, with an average benefit of 31 $MtCO_2e$ per year, over the next 100 years. However, if one subtracts their figures for fossil fuel displacement and building substitution benefits, the figure would be just 2.4 $MtCO_2e$ per year. Lippke et al. [50] illustrated the supposed

benefits of product substitution over a long period, with accumulated substitution and displacement benefits reaching around 500 tonnes of carbon per hectare after 160 years (and, in another case, around 1100 tonnes after 300 years). This dwarfs both the forest carbon (which peaks at around 160 tC/ha at the end of each rotation) and the HWP carbon pool itself. Oliver et al. [51] also relied heavily on substitution benefits to reach their conclusion that increased use of wood products in buildings and infrastructure could result in global reductions in $CO_2$e emissions of 14–31%.

Hill [52] noted the different approaches that can be applied to modelling loss of carbon from the HWP pool, and Law et al. [42] made more conservative assumptions about this and reached different conclusions to those covered in the previous paragraph. Referring to forestry in Oregon, they concluded that rotation periods should be increased to facilitate greater carbon storage in the forest, and that harvest residues should not be utilised for bioenergy.

Moving the focus from the forest towards the buildings, a dynamic stock model of the residential building sector in Austria shows that, with a significant increase in the timber share of the market, Austrian buildings can triple the carbon stored by 2100, without Austria losing its status as a net exporter of timber [53]. For Switzerland, Heeren and Hellweg [54] modelled the volumetric flows of materials through the residential construction industry. In their increased-wood scenario, the total stock of wood increases from 31 Mt in 2015 to 46 Mt in 2055 (out of a total material stock of approximately 1330 Mt). A material flow analysis (MFA) of wood in the construction sector in Europe, with a focus on engineered wood products, shows potential for 46 $MtCO_2$e p.a. carbon storage in HWP by 2030 [55]: this is approximately fifteen times the CCC assessment for the 2050 potential of the UK residential sector, as discussed in the introduction. Pittau et al. [56] assessed the potential of retrofit of buildings across the European Union (EU) as a carbon sink. Their analysis picks up on the case for rapid action, i.e., by incorporating fast-growing crops into building facades.

### 3.2.4. Greenhouse Gas Fluxes in Forestry

The PCR for wood-based construction products, EN 16485:2014 [57], draws a line between what might be termed the technosphere and the ecosphere—a 'system boundary with nature.' On the technical side of the boundary lie the anthropogenic inputs and processes leading to the production of the timber; the sequestration of carbon from the atmosphere into the crop is included, as this is a physical property of the product. All other forestry processes are understood to be part of nature, even in a commercial monoculture. Therefore, other carbon and GHG fluxes between the atmosphere, the trees, the soil, and the groundwater associated with the growth and decay of trees are outside the scope of the PCR and of standard LCA practice. This simplification is both contestable and potentially justifiable. On one hand, the GHG fluxes in a commercial forestry stand are different from those that would occur if the land were put to different use. To the extent that increased demand for construction timber supports either new afforestation or future rotations of existing forestry, then the product should clearly account for its contribution to the net change in emissions. On the other hand, it is unlikely that sufficient data is available to characterise the full range of GHG fluxes in the range of possible contexts (soil type, tree species, climate, stand age, and rotation number, to name but a few variables) to make inclusion in LCA practicable. If such emissions can be shown to be only a relatively minor part of the 'complete' GHG account for timber products, then that would add to the justification for leaving it out of scope.

The UK Government's reporting on land use and forestry emissions [58], in accordance with the relevant Guidelines for National Greenhouse Gas Inventories [59], provide a top-down annual perspective. Figures for 2017 show forest land acting as a net carbon sink of 18 $MtCO_2$e, with a further 2 $MtCO_2$e incorporated in harvested timber. Set against this, the combined effect of fire (including associated methane emissions), direct nitrous oxide emissions, and drainage of organic soils shows losses of 0.18 $MtCO_2$e, which is equivalent to nearly 10% of the biogenic carbon in harvested timber.

A review of the role of UK forests in combatting climate change includes studies focusing on methane and nitrous oxide emissions from forest soils, and numbers span a wide range, from a small

sink right up to emissions of 1450 $kgCO_2e$ $ha^{-1}yr^{-1}$ for standing forest [60]. Those who have attempted to include soil carbon in LCAs agree that it is potentially significant, but not on whether it is a positive or negative contribution [61]. Given that forestry can produce sawn softwood at a rate of around 4 $m^3ha^{-1}yr^{-1}$ [62], such emissions would be significant in the context of construction timber LCAs. In general, nitrous oxide emissions are higher from relatively warm soils with a low carbon-to-nitrogen ratio and high application of nitrogen fertilizer [60], none of which applies to upland forestry in the UK, but the picture may be different for European forests supplying UK markets.

Another source of emissions relates to loss of soil carbon when land is drained for afforestation, and after any forestry rotation when the decision is taken to remove the stumps, roots, and brash for use as fuel. This involves disturbing the soil to a depth of about one metre which, in organic soils, results in significant levels of oxidation of soil carbon: a range of 14–20 $tCO_2$ $ha^{-1}yr^{-1}$ has been suggested for the first few years of the second rotation, but that by the end of the second rotation, the soil will have recovered the earlier losses [60]. Law et al. [42] suggest that the utilisation of harvest residues for bioenergy results in increased emissions. Others suggest that, at the landscape level, the impact of stump harvesting on soil carbon has been overstated, with losses peaking at less than 4 $tCO_2ha^{-1}yr^{-1}$ a few years after the harvest [63]. In their review of the environmental impacts of stump harvesting, Walmsley and Godbold [64] indicate rewards of an additional 100 to 250 MWh per hectare for those who harvest stumps for bioenergy. The corresponding substitution benefit in relation to natural gas use would be a one-time benefit of approximately 20 to 50 $tCO_2e$, which would need to be set against the soil organic carbon losses referred to above. Who takes responsibility for such emissions—is it the biomass energy facility that takes the roots, the forester who elected to provide them when they could have been left in the ground, or the wood product industry which demanded the harvest that made the roots available? Currently, if it rests anywhere, responsibility rests with the forester, for the reason given at the start of this section, but we would suggest that all actors are part of the same system, so responsibility should be allocated between them.

Nothing in this section should be understood as a blanket rejection of a sustainable HWP supply chain: it is more a case of drawing attention to challenges. On the other hand, regarding stump harvesting, soils with less than 5 cm of peat have been identified as 'low risk' [65], and Ortiz et al. [63] concluded that stump bioenergy delivers climate change mitigation (compared to natural gas) after 12–28 years.

### 3.2.5. Competition for Resources

Although LCA can be used to investigate direct and indirect land use change, its capacity to offer a complete quantitative assessment, including socioeconomic aspects of land use, is debatable [66]. Furthermore, the methods discussed above have little to say on the capacity of the biological cycle to deliver the materials we might require for the construction industry, given competing demands on the land, such as biodiversity, agriculture, and urbanisation. The construction sector is not the only one looking towards the biological cycle to reduce its carbon footprint. Others include biomass power generation, biofuels for transport including aviation, and bioplastics where once the primary focus was on biodegradable plastics, but growth is now driven by the like-for-like substitution of petroleum-based products with biogenic equivalents [67]. Analysis of competition for land at a global level indicates pressure on forests to give way to food production [68]. On the other hand, Rounsevell and Reay [69] indicate a decrease in land area dedicated to food production and therefore offer a more sanguine picture about the possibility of some parts of the world contributing additional timber to the markets.

A sharp increase in demand in producing countries could have significant implications for net importers of construction timber, such as the UK, and for land use in general. The 2030 scenario in Hildebrandt et al. [55] requires additional land of approximately 7 MHa, which is equivalent to the areas that are forested and available for wood supply in Austria, Latvia, and the UK combined [70]. Allwood [71] states that global competition for biomass makes the general replacement of steel with

timber unlikely, but Ramage et al. [62] imply that, in Europe, at least, this level of ambition would be justifiable: they estimate that no more than 30% of Europe's existing forest area would be needed to keep the population of the continent housed in timber.

One rare study that combines a construction-related LCA with a look at the capacity of the local industry to deliver the material required—cork insulation—does not ask about the consequences of increasing the demand and supply of the material in the long run [72]. Such consequences might include land use change and increasing prices pushing existing cork users towards synthetic products. This is where consequential LCA can help, and—it has been argued—might even be essential to socially responsible decision-making [73]. This, of course, opens up a further set of challenges: in their study of how methodological choices affect the results of consequential LCA in this context, De Rosa et al. [74] tested one case study (structural spruce timber grown in the south of Sweden) with eight different sets of methodological choices, and found that they gave widely divergent results, ranging from net sequestration of 24 $kgCO_2e/m^3$ of timber to net emissions of 3220 $kgCO_2e/m^3$.

Competition for land and natural resources will remain fierce in the coming decades given current and projected consumption levels, as well as urbanisation and population growth trends. Future studies aiming to quantify the potential for, and benefits of, a global uptake of timber in construction should bear these competing demands in mind and work within realistic assumptions on resource constraints.

## 4. Conclusions

This paper looked into unanswered questions and future challenges linked to an increased use of timber in construction. Although LCAs of buildings and structures consistently support the view that use of timber results in a building with lower embodied carbon, this is not the full story. Confronting the challenges associated with LCAs of timber products identified in this paper is unlikely to overthrow this general conclusion, but a better understanding of the issues involved can inform better policy-making and better management choices throughout the value chain, from forest management, to building design, to demolition and material recovery.

Within the scope of the LCA itself there are difficult questions about the potential impacts at end of life that are rarely answered, and factors potentially leading to changes in GHG emissions are neglected by LCA altogether.

To understand the life cycle impacts of timber in construction, detailed consideration of the end-of-life scenarios is required. A limited selection of scenarios is typically offered, without justifying the assumptions behind them or the relative likelihood of those scenarios being applicable. Examples include: future landfill availability; biodegradation and landfill gas production rates in the local conditions; energy recovery efficiency; and carbon intensity of future counterfactuals (e.g., electricity produced from a grid that will be largely decarbonised when EoL is reached).

Looking beyond current LCA practice, a body of research exists into the impact of the timing of carbon emissions in LCA. This points the way towards identifying a credible method for allocating some credit to harvested wood products for the temporary carbon storage service provided, but as yet there is no consensus on how to apply these dynamic approaches.

Another impact of using more timber in construction that needs consideration concerns the land use requirement for increased timber production. Firstly, emissions associated with natural processes of growth and decay in the forest are not included in LCA: the significance of this needs to be better understood, together with more information on how to manage different types of land over long time periods to optimise GHG fluxes. Secondly, there is the question of whether sufficient land is available. There is some evidence that Europe, for instance, could meet demand for rapidly growing timber use from its own forests. However, the consequences of meeting increased demand will inevitably include displacing some existing consumers from the market for home-grown timber and potentially towards alternatives with higher life cycle impacts, such as tropical timber or synthetic materials. Therefore, any temptation to use more timber than is necessary for the job (e.g., heavy engineered timber elements in preference to a lighter timber-frame construction where substitution benefits are not enough

to justify the choice) should be resisted. In a growing global economy, extracting more resources than necessary—renewable or not—will never be consistent with the principles of sustainability. Furthermore, what applies in Europe does not necessarily apply in other parts of the world—the tropics being a case in point, where deforestation is in charge and further demand for timber undesirable.

All the issues above represent fundamental avenues of research to inform the ongoing debate over increased use of timber in buildings, and although, from a life cycle perspective, sustainably produced timber is likely to be preferable to nonrenewable materials in many contexts, the selection of more sustainable materials does not absolve us from the need for sustainable consumption of those materials. This involves reckoning with questions about efficiency, utility, and durability and—probably the most important question of all—whether to consume at all.

**Author Contributions:** Research and primary authorship: J.H.; Advice, additional research, and review: F.P. All authors have read and agreed to the published version of the manuscript.

**Funding:** The research presented in this article has been funded by the UK's Engineering and Physical Sciences Research Council (EPSRC)—Grant Agreement No. EP/R01468X/1.

**Conflicts of Interest:** The authors declare no conflict of interest.

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
