# Peer review of "More Timber in Construction: Unanswered Questions and Future Challenges"

_sustainability, doi:10.3390/su12083473_

Round 1

Reviewer 1 Report

Dear author,

The paper presents a well written view of current challenges and questions arising from the use of timber product in construction. The paper presents its findings in a well structured way and captures its scientific base from a decent overview of the current state of art.

Only some very minor adjustements with regard to spelling are suggested:

Line 52: MtCO2 à Mt CO2

Line 64: plant à planet

Linz 414: there is unlikely to be sufficient data available à this sounds very weird. Maybe change to; it is unlikely that sufficient data is available

Kind regards

Author Response

We thank the reviewer for the useful comments - which we have addressed in full in the revised manuscript

Reviewer 2 Report

The paper addresses the interesting topic of the environmental loads and benefits associated with the end of life options for timber used in building construction.

The paper is a review and it is well written.

Different studies and approaches to evaluate the LCA of timber buildings are compared.

The bibliography is wide and complete.

The methodology is appropriate.

The conclusions aren’t duly defined. Infact, it is not clear to what extent the increase in the use of timber in building construction is sustainable or not.

Author Response

We thank the reviewer for the positive and constructive comment. In response to the one on the conclusions we have expanded a part specifying what the evidence says in terms of the sustainability of timber in construction although the scope of the paper is more on highlighting existing issues rather than answering that question in a definite way.